# Barriers and Facilitators of Physical Activity in Pregnancy and Postpartum Among Iranian Women: A Scoping Review

**DOI:** 10.3390/healthcare12232416

**Published:** 2024-12-02

**Authors:** Linda E. May, Sarah J. Moss, Anna Szumilewicz, Rita Santos-Rocha, Najmeh A. Shojaeian

**Affiliations:** 1Department of Kinesiology, College of Health and Human Performance, East Carolina University, Ward Sports Medicine Building, 371A, Greenville, NC 27834, USA; 2Physical Activity, Sport and Recreation Research Focus Area, Faculty of Health Sciences, North-West University, Potchefstroom 2531, South Africa; 3Department of Fitness, Faculty of Physical Culture, Gdansk University of Physical Education and Sport, 80-336 Gdansk, Poland; 4ESDRM—Sport Sciences School of Rio Maior, Santarém Polytechnic University, 2040-413 Rio Maior, Portugal; 5SPRINT—Sport Physical Activity and Health Research & Innovation Center, 2040-413 Rio Maior, Portugal; 6Department of Sport Sciences, Faculty of Humanities, Bojnourd Branch, Islamic Azad University, Bojnourd 9417697796, Iran

**Keywords:** physical activity, barrier, facilitator, pregnancy, postpartum, Iranian women

## Abstract

Inactivity during pregnancy and postpartum is largely a result of women’s attitudes and misunderstandings of physical activity, especially in Iran. This scoping review critically assesses the barriers and facilitators influencing physical activity among pregnant and postpartum Iranian women to provide the basis for future physical activity interventions. Ten databases and platforms were searched up to 1 June 2024: Medline, SportDISCUS, PsycINFO, EMBASE, CINAHL, Cochrane Review Database, Clinical Trial, SID, ISC, and Web of Science. Grey literature sources were included to retrieve original publications on barriers and facilitators during pregnancy and postpartum among Iranian women. The search resulted in 2470 identified studies screened for inclusion criteria. After screening both abstracts and full texts, 33 of the studies were included, and data were extracted and charted. Findings were summarized in alignment with the objectives. The results show that the basic physical activity barriers are intrapersonal, interpersonal, and environmental factors. Facilitating factors include using E-learning resources and combined interventions to educate women and provide awareness of the existence of exercise classes. Social and emotional support by family members and other women in the same situation can be effective. Overall, the study of obstacles to and enablers of physical activity during pregnancy and postpartum is ongoing. In addition to highlighting the present situation in Iran, this study identifies further opportunities for future research on the development of appropriate interventions to reduce the barriers and strengthen the facilitators for physical activity among pregnant and postpartum Iranian women with trained groups, including skilled healthcare providers.

## 1. Introduction

Many guidelines recommend that regular physical activity (PA) has several health benefits for mothers and their children during pregnancy and postpartum [1,2,3]. However, women report declining PA levels as pregnancy progresses [4,5]. Several studies report evidence demonstrating that PA can prevent birth complications for mothers and infants [1,3,6]. For instance, PA during pregnancy decreases the risk of placenta previa, hypertension, gestational diabetes, and cesarean section [7]; improves maternal glucose levels [8]; supports healthy neonatal birth weight [6]; improves the quality of life (health, physical comfort, mental and social dimensions) [5]; develops the motor and social skills in infancy [9]; and leads to postpartum benefits [10,11]. Moreover, PA throughout gestation is associated with postpartum weight loss, higher scores on psychosocial health measures [6,11], improved sleep [1], and reduced postpartum depression [12]. Considering these outcomes and the beneficial effects of PA during pregnancy on birth outcomes [13,14], it is advantageous for every woman to engage in exercise; however, the rate of PA decreases during pregnancy and postpartum [1,4,5,6].

In a comprehensive study conducted in Iran, the patterns of PA domains, insufficient PA, the intensity of PA, and sedentary behavior at both the national and provincial levels were assessed in 2021. The findings revealed that 49.42% of males and 51.95% of females had an age-standardized prevalence of a sedentary lifestyle in daily life [15]. Moreover, it was reported that active women tended to reduce their PA levels during pregnancy, and this pattern could persist into the postpartum period [16]. Abedzadeh et al. stated that only 22% of pregnant women were aware of the benefits of exercise during pregnancy [17]. The study by Esmailzadeh et al. showed that 70% of pregnant women did not participate in any PA [4], and Bahadoran et al. reported that 98% of individuals engaged in light-intensity PA, while less than 2% participated in moderate-intensity PA [5]. In a study in Tehran, 52% of the pregnant women in the first trimester did not participate in any exercise, 44% of this population participated in low-intensity, and 3.6% did moderate-intensity PA. In the second trimester, 70% of women did not engage in PA, while only 28.9% performed low-intensity and 0.4% achieved moderate-intensity PA. Also, within three months after delivery, 79.6% of postpartum women did not achieve the PA guidelines. Among 21.4% of active women, 18.2% performed low-intensity and 2.2% participated in moderate-intensity PA [4]. From the limited research available, several reasons have been reported to affect an individual’s ability to meet exercise recommendations [18]. Although the literature reports some barriers to participating in PA for women during pregnancy and postpartum around the world [16,18,19], there are also special factors that affect the PA level among Iranian women. These factors include intrapersonal struggles [20,21,22], interpersonal constraints [23,24,25], and environmental opportunities [20,26].

The focus of this study on Iranian women is particularly relevant given the cultural and social norms that influence health behaviors, including physical activity, in Iran [4,5,17]. These unique barriers, such as cultural restrictions on physical activity [27] and limited access to resources [23], necessitate targeted interventions. Furthermore, Iran is experiencing rising rates of pregnancy-related health issues, such as gestational diabetes and hypertension [28], making it crucial to address PA during these periods to prevent long-term health consequences. Additionally, aligning with national health goals to reduce maternal health disparities [29], this study aims to fill the gap in the literature by identifying the barriers and facilitators of PA, supporting efforts to maintain or increase PA levels, and using the findings to guide future exercise interventions for pregnant and postpartum women.

## 2. Materials and Methods

This scoping review was conducted based on the protocol by Levac, Colquhoun, and O’Brien (2010) in five stages [30]. The review was guided by the Preferred Reporting Items for Systematic Reviews and Meta-Analyses extension for Scoping Reviews (PRISMA-ScR).

### 2.1. First Stage: Aim and Research Questions

This scoping review aimed (1) to identify the barriers and facilitators of PA during pregnancy and postpartum among Iranian women, (2) to support maintaining or increasing PA levels in this population, as well as (3) to use the study outcomes to develop future exercise interventions during pregnancy and postpartum.

### 2.2. Second Step: Relevant Studies Identified

All qualitative, quantitative, and mixed-methods studies as well as systematic and non-systematic reviews published in English or Persian were eligible. Gray literature was also searched. The search included information up to June 2024. Ten electronic databases were consulted, including Medline (n = 10), SPORTDiscus (n = 80), PsycINFO (n = 2), EMBASE (n = 174), CINAHL (n = 10), Cochrane Library (n = 3), Clinical Trials (n = 1370), SID (n = 703), ISC (n = 113), and Web of Science (n = 5). The following keywords and MeSH terms were used in the searches: “physical activity, exercise, facilitators, barriers, pregnancy and postpartum, Iranian women”. Databases were searched for titles, abstracts, and keywords containing the terms “women”, “facilitator”, and “barrier”, as recommended for effective search criteria in scoping reviews [31]. A comprehensive search strategy was developed combining the following keywords: [(barriers OR constraints), (facilitator OR enabler), (physical activity OR exercise OR motor activity), (pregnancy OR pregnant women OR antenatal OR prenatal) AND (postpartum period OR postnatal)]. Inclusion and exclusion criteria (see Table 1) were applied during the selection process for eligibility. A full electronic search strategy can be found in Table 2 and Table 3 and Appendix A.

### 2.3. Third Step: Study Selection

After entering the searched articles into the Mendeley software (version 1.19.8) and removing duplicates, the screening process was performed by examining the title and abstract of the articles. Then, the full texts were examined by authors separately based on the inclusion and exclusion criteria. If there was a difference of opinion, then it was resolved through discussion and exchange of viewpoints. All studies adhering to the inclusion criteria were first screened according to the titles, then the abstracts and full texts were reviewed for the included articles. The study selection process and reason for exclusion are presented in a PRISMA flow diagram (Figure 1).

### 2.4. Fourth Step: Charting the Data

A data extraction template was constructed using Microsoft Excel to extract the “barriers and facilities of PA” information stated within each included study (see Appendix A). The following information was extracted in addition to the barriers and facilitators, number of participants, region, population characteristics, concept, and context of the study.

### 2.5. Fifth Step: Collating, Summarizing, and Reporting Results

The data extracted from the publications were collated and summarized based on the study design, participant characteristics, physical activity features, and findings reported. The summary provided insights into the nature and distribution of the included studies. To identify themes, a thematic analysis was conducted on the extracted data, focusing specifically on barriers to and facilitators of physical activity during pregnancy and postpartum. The key findings from each study were reviewed and categorized into sub-themes related to each of the two main themes (barriers and facilitators).

Researchers independently reviewed each study, and their findings were discussed in consensus meetings to ensure accuracy and consistency in identifying the themes. This iterative process allowed for the refinement and grouping of the data into overarching themes that addressed the research questions of the review. As part of this process, the key findings from Table 2 were grouped into categories that reflected the central factors influencing physical activity during pregnancy and postpartum. These categories were further distilled into themes that directly addressed the barriers and facilitators identified across the studies. The sub-themes were further refined to create categories and sub-categories.

To enhance transparency and clarity, we quantified the prevalence of each theme. For example, 51% of the studies (n = 17) identified lack of knowledge about exercise programs as a major barrier to physical activity during pregnancy, while 49% of the studies (n = 16) reported social and emotional support as a facilitator. Other barriers, such as cultural restrictions, and facilitators, such as healthcare provider support, were similarly categorized and quantified based on their frequency across studies.

The themes presented in Table 3 and Figure 2 were derived from this analysis and represent the central factors influencing physical activity during pregnancy and postpartum among Iranian women. Figure 2 provides a visual summary of these key themes and sub-themes, directly informed by the thematic analysis process described above. This figure categorizes the factors under six main headings: guidelines on physical activity, prevalence of activity levels, barriers to activity, enablers of activity, strategies to promote activity, and recommendations for future research. Each of these headings encompasses specific sub-themes, highlighting the factors that play a crucial role in shaping physical activity behaviors during pregnancy and postpartum.

#### Consultation

People other than the authors were not involved in this review’s design, reporting, or dissemination.

## 3. Results

### 3.1. Characteristics of the Studies

A total of 2470 studies were identified for screening (pregnancy n = 1805, postpartum n = 563), with 33 studies included in the final analysis (see Table 2). The studies included a variety of designs, including descriptive methods, randomized clinical trials, and qualitative research. Figure 1 presents a study selection flowchart, with a summary of included studies detailed in the Appendix A. Also, the narrative summary of findings is organized according to our research questions (Table 2). Two major themes followed by seven subthemes, 16 categories, and 40 subcategories were developed in this study (Table 3).

We included 24 studies with a total of 4217 pregnant women [17,20,23,24,26,27,29,32,33,35,36,37,39,41,42,43,44,46,48,49,50,52,54,55], 54 care providers [26,37,51], and 1890 postpartum women [4,11,21,22,25,34,40,45,47] ages 18 to 45 years old. Most pregnant women were recruited between the second and third trimesters [20,24,26,27,32,33,35,43,44,50]. Meanwhile, six papers were started in the first trimester [29,38,39,42,46,48].

Some inclusion criteria for pregnant women were as follows: Iranian women with a healthy or low-risk pregnancy, nulliparous or second pregnancy, and singleton [17,23,26,27,32,41,42,44,46,48,52]. The inclusion criteria for postpartum women included being healthy and primiparous [4,11,22,34,40,45,47]. All women had no restriction on PA. In five papers, the study participants had a BMI ≥ 25 kg/m^2^ [26,40,43,46,47,52], and one study had participants with gestational diabetes mellitus (GDM) [37]. All postpartum women participated between 6 months and 1 year after delivery [4,11,21,22,25,34,40,45].

Regarding the barriers and facilitators of PA during pregnancy and postpartum, datasets were derived from in-depth interviews [25,26,33,36,37,38], questionnaires [4,11,17,20,21,22,24,25,26,27,29,32,33,34,36,39,40,41,42,43,44,45,46,48,50,51,52,55], open-ended questions [23,26,43,44], and semi-structured interviews [32,37,47,49].

The quality of the studies included in this review was assessed based on the clarity of the research design, sampling methods, data collection and reporting procedures, and the validity and reliability of the measure instruments used. While many studies included validated questionnaires [11,20,22,23,43,48], some did not provide detailed information about the psychometric properties of the tools used [17,25,42,52]. Additionally, a variety of study designs were employed, including randomized controlled trial [29,34,35], cross-sectional studies [17,46,48], quasi-experimental designs [20,42,43], and qualitative research [25,26,40]. Despite this variation, all studies attempted to address the key factors affecting physical activity during pregnancy and postpartum. However, the heterogeneity of these studies in terms of design, measurement tools, and reporting practices may limit the ability to directly compare findings.

Several limitations of the retrieved studies should be acknowledged. Common limitations in the retrieved studies included small sample sizes [17,21,26,33,39,41,47], which limited the generalizability of findings. Moreover, several studies were conducted in specific geographic regions, such as urban centers, which limited the applicability of the findings to broader populations, including those in rural areas or from different racial or ethnic backgrounds [17,22,23,33,39,41,43,46]. Furthermore, many studies relied on self-reported data [11,21,27,39,41,51], which may introduce recall bias or inaccuracies in the reporting of physical activity levels. Another limitation was that some studies did not consider the attitudes, perceptions, and behaviors of husbands and healthcare providers (e.g., physicians and midwives), which could have provided valuable insights into the reasons for physical inactivity among pregnant and postpartum women [21,37,44,55]. This exclusion could have led to an incomplete understanding of the social and cultural factors influencing physical activity levels among pregnant and postpartum women. These factors, if considered, may have provided valuable insights into why some women engage in physical activity while others do not.

### 3.2. What Barriers Affect the Physical Activity Level in Pregnancy and Postpartum?

In this study, eight papers (28%) reported the barriers [11,24,25,32,38,47,48,51], 13 studies (39%) showed facilitators [20,22,26,29,33,34,35,36,39,40,43,49,50,55], and 12 papers (37%) mentioned both facilitators and barriers [4,17,21,26,27,37,41,42,44,45,46,52]. Generally, women reported numerous barriers to PA. Three themes were identified from the content of PA barriers: intrapersonal factors [32,46], interpersonal factors [27,37], and the environment [47,51]. More details are discussed in the following text.

#### 3.2.1. Intrapersonal Factors

Intrapersonal factors (derived from individuals) were dichotomized into health-related and non-health-related categories [56]. Examples of health-related barriers included physical limitations [48,52], health conditions [4,32,46], tiredness, dizziness, and fatigue [26,32,47,48]. Non-health-related barriers were related to psychosocial attitudes toward PA, a specific lack of motivation [19,26], or a lack of self-efficacy [11,22,36,39,41,45].

Despite participants usually being aware of the current guidelines for PA for non-pregnant adults, they were not aware of separate guidelines for PA during pregnancy [16] or postpartum [19]. In 17 studies, a lack of knowledge regarding exercise classes and educational programs for pregnant women in hospitals, health centers, or sports facilities was reported to affect the rate of PA participation [17,21,23,24,25,26,29,32,37,39,40,41,42,43,44,48,52]. Additionally, these studies stated that these opportunities for pregnant women are limited in developing countries [24,26,37]. Two studies indicated motivation as an important factor for exercising during pregnancy and postpartum [26,44]. In an in-depth interview, a few women who felt at risk of pregnancy-induced complications tried prenatal exercise classes, for example, women with a history of familial diseases or high blood pressure [26]. Physiological and physical conditions and heaviness were noted in nine papers [4,21,23,26,32,44,46,48,52]. For instance, the morphological changes experienced during pregnancy were an important reason for limiting PA. As pregnancy progresses, the growing abdomen makes it difficult to carry out certain activities, such as running. Consequently, pregnant women often feel that they should reduce the amount and type of exercise during the advanced stages of pregnancy [4,5]. Another key barrier is physiological changes. Shortness of breath and fatigue are important obstacles among postpartum women that prevent them from being active [4,16,32,47]. A survey in Tehran indicated that 68% of pregnant, and 49.9% of postpartum women felt fatigue [4].

Five papers directly mentioned that pre-pregnancy or early pregnancy body mass index (BMI), and exercise habits before pregnancy had a significant relationship with the total PA intensity [21,22,39,46,48]. Specifically, women with a BMI above 30 kg/m2 before or at the beginning of pregnancy had a higher mean energy expenditure and intensity [46,48]. Additionally, gestational age had a significant correlation with energy expenditure. Two studies found that engagement in PA levels is higher during the first trimester of pregnancy compared to the second and third trimesters [39,46].

#### 3.2.2. Interpersonal Factors

The interpersonal factors (elements that affect how people interact with each other) are included in social interaction [25,36,42,44,46], social support [19,20,22,25,36,41,44,46,47], and cultural and religious beliefs prevailing within the society [27,32,36,37]. While normative beliefs or subjective norms are derived from an intrapersonal level, for this review, they were examined at the interpersonal level, as they describe persons identified as valuable for informational and motivational support for PA during pregnancy.

In 17 papers, the role of social support was mentioned [5,20,21,23,26,27,32,36,37,40,42,44,46,47,48,52,55]. Pregnancy is a critical period for family members, and the role of spouses is considerable. These studies in Iran examined the barriers and facilitators of Iranian men’s involvement in perinatal care. The findings showed that gender authoritarian attitudes (including subjective norms, stereotypes, and hidden fears), constraints (including individual, organizational, socio-economic, and legislative constraints), and incentives (including individual, family, economic, legislative, and organizational incentives) are the main factors that men face which affect maternal and neonatal health promotion programs [53,57,58]. For example, it was noted that some husbands did not permit their wives to engage in PA because they believed it to be dangerous for their babies [5,26].

Additionally, family discouragement was perceived as a great barrier to PA [32,46]. In some Iranian regions, it is still common for couples to live with the husband’s family. Therefore, the effects of family attitudes and behaviors, especially during pregnancy, are more pronounced. Pregnant women mentioned that the bustling atmosphere of the family and a lack of understanding of their conditions led to increased stress [32,37]. Through in-depth interviews with pregnant women, it was revealed that, despite their belief that the presence of other women in pregnancy classes can motivate them to attend [44], many sociocultural norms prevented them from offering mutual support [26,32]. In seven studies, a lack of communication (role of healthcare and physicians, other pregnant people) was discussed [4,26,32,37,44,47]. The results of a questionnaire completed by postpartum women indicated that physicians (91.6%) and midwives (84.4%) had a significant influence on physical activity (PA) participation. However, studies showed that only 62% of women consulted with a physician. Furthermore, health centers play an effective role in the postpartum period, with physicians (88.9%) and midwives (82.2%) significantly influencing PA participation [4].

The roles of pregnant and postpartum women as child caregivers, along with domestic responsibilities, lead to a lack of time and energy to exercise regularly [24,38]. Ahmadi et al. (2021) found that the number of children in the household negatively impacts mothers’ PA. Women with a higher number of children expended more energy by carrying out light and moderate household/caregiving responsibilities. In contrast, childless pregnant women had a higher energy expenditure for carrying out occupational, sports, and vigorous activities [46]. However, a study in Southeast Asia found that the number of children in the household did not impact maternal PA levels. Indonesian mothers who had one child did not differ in PA levels and sitting times compared to mothers who had more than one child in their household [19].

In 15 studies, cultural and social beliefs and attitudes were revealed [4,17,21,24,26,32,33,36,37,40,43,44,46,48,52]. Bahadoran et al. (2015) claimed that differences in cultural, social, and religious beliefs can be effective in increasing the amount and the mode of PA [5]. Structural problems, such as husband’s opposition and the affordability of classes, affected their ability to attend classes and be more physically active [19,26,27,37,48,53]. In developing countries, culture is affected by religion and beliefs [26]. Although the Islamic religion emphasizes the importance of PA and demands an active lifestyle, it is difficult for women to be physically active in this strict culture. One aspect of the Islamic faith is the expectation for modest dress, including clothing that covers most of the body [18,46]. It was reported that dress and negative perceptions toward Arab Muslim women who are involved in sports or exercise are the main barriers to participation in PA. Among the participants, 36.3% strongly agreed that the lack of female facilities was a barrier to being active. Additionally, most women felt stressed (76.1%), embarrassed to exercise in public (59.8%), and uncomfortable wearing gym clothes (82.7%). The majority of women had high levels of religious affiliation and reported that their religious faith influenced their choices, personality, and behavior [18].

#### 3.2.3. Environmental Factors

There are a variety of environmental factors and socioeconomic restrictions. These are composed of two categories: physical factors (lack of health and safety principles in sports environments, long distance to PA facilities) and organizational and structural factors associated with sports environments (costs, facilities, equipment, and location) [32,34,47,48].

In nine studies, a lack of equipment and sports facilities were mentioned [23,26,32,40,44,46,47,48,51]. Ahmadi et al. (2021) stated that the location affects the participation rate in PA. Although all participants in the study lived in the city, it is noted that PA promotion may need tailoring to address the needs of women living in both rural and urban settings, particularly since women in urban areas generally have greater access to sports facilities (such as clubs) than women in rural areas [46]. However, urban areas are often not suitable for walking or other activities due to air pollution and crowded, built-up environments [46,48]. In a study examining the relationship between intensity, barriers, and correlates of PA among Iranian pregnant women in their second trimester, the findings revealed that women who did not attend childbirth preparation classes expended more energy (MET-hour/week). Contrary to expectations, these classes in Iran do not provide training on the importance and safety of PA during pregnancy [46]. In a study conducted on midwifery personnel at health centers in Isfahan city, it was found that all research units exhibited poor performance (100%), and most individuals had only average awareness (50%) regarding exercise training during pregnancy [51]. Therefore, there are significant opportunities to integrate PA promotion into childbirth preparation classes.

Several studies on pregnant women noted that homes are small and not suitable for PA. Some pregnant women believed that completing chores like washing and cleaning was adequate exercise [26,48]. The economic situation and financial problems were other barriers to participation in PA classes in six studies [23,26,27,36,44,46]. Some participants stated that they preferred to participate in a sports class that was not expensive and that had a private trainer. Furthermore, most pregnant women were unaware of free programs [35,46]. According to the study by Ahmadi, a higher income was also associated with increased energy expenditure, and lower income levels have similarly been associated with decreased PA; the researchers suggested that those with lower incomes should be targeted in the future promotion of PA [46]. It seems that there is a relationship between income, job, and rate of PA. Pregnant and postpartum women spend a significantly greater amount of time sitting compared to being physically active. The time spent sitting might be related to their employment status. Increased technology has led to changes in employees’ lifestyles, with prolonged desk-based work and reduced activity [19].

### 3.3. What Facilitators Affect the Physical Activity Level in Pregnancy and Postpartum?

A survey in Iran showed that exercise during pregnancy and postpartum elicited a sense of happiness and enjoyment among women (94.7% and 94.2%, respectively) [4]. Therefore, it seems that some factors could help women to improve their PA. The findings of Kianfard’s research (2021) indicated that maternal health is not the primary motivation for implementing exercise programs during pregnancy [26]. Related to this issue, providing a bonus for exercising (e.g., earning points toward gifts) might help motivate pregnant women to exercise regularly [19]. Some women claimed that they did not have enough money or time to attend exercise classes. Employing a variety of strategies, including having mobility at home [26,32,35], access to facilities [16,48], and awareness of free classes [23,37], were identified as essential motivators for PA.

Social and emotional support from husbands or partners, family, and friends is necessary [5,20,21,23,26,27,32,36,37,40,42,44,46,48,52,55]. In five papers, the role of family members was mentioned as a facilitator [21,23,32,36,55]. For example, some women’s husbands gave general encouragement, whereas others accompanied women while walking or exercising in gymnasiums or parks [21,23]. Social networks and the community [16,23,32] played an important role in making regular exercise possible. Women described neighborhoods where they felt free to walk, even at night, and where there were places for children to play [59].

Educational methods and interventions were suggested by 12 papers [20,25,26,29,32,33,34,37,42,45,49,50]. Combined exercise intervention methods, such as face-to-face plus monitored home programs, have a lower cost and could improve PA levels [35]. Women also said that providing guidance or education about how to safely do exercises during pregnancy and postpartum through different methods (including face-to-face, phone, and SMS counseling and a booklet) would make it easier for them to be physically active [22,24,25,34,50,59]. This means that healthcare providers must provide all information about the existence of these classes to pregnant women, and social networks and media must inform them as well [5,17,26,27,37,48,52]. Many studies stated that online training and E-learning applications could increase antenatal and postnatal PA [20,26,34,49,60]. E-learning was described as having a significant effect on training factors (the role of people who communicate with pregnant women), enabling factors (including appropriate places) [23], perceptual factors (awareness of type, intensity, frequency of exercise, reasons for doing exercises), and attitude (social and individual attitude) [43] that increase PA among women.

In a study by Ahmadi, a mobile application (app) was developed to promote PA among pregnant women during the COVID-19 pandemic. The app was designed based on a needs assessment with pregnant women, childbirth preparation teachers, and health experts, ensuring the content met their specific needs. The app included multimedia elements (text, photos, videos, GIFs) and covered 12 domains, such as the benefits of PA, safe exercises (e.g., walking, yoga), daily activities, posture, and relaxation exercises. It also addressed perceived barriers, social support, and enjoyment. The app was provided to the intervention group with training on usage and weekly reminders through a national messaging platform. Data from pre- and post-intervention questionnaires showed significant improvements in perceived benefits, enjoyment, and overall PA levels in the intervention group compared to the control group. The study demonstrated that mobile apps can effectively promote PA among pregnant women, especially when access to in-person classes is limited, and suggested integrating such apps with face-to-face sessions in health centers for better results [20]. Another study employed a mixed-methods design (both quantitative and qualitative) to investigate the impact of an E-learning intervention on PA levels among pregnant women. The first phase involved a comprehensive literature review followed by semi-organized interviews with pregnant women. This stage utilized qualitative methods to capture the perspectives and experiences of the participants, providing a deep understanding of the barriers to and facilitators of PA during pregnancy [26].

The second stage involved the implementation of an E-learning program intervention based on the insights obtained from the first stage. A randomized controlled trial (RCT) design was employed, with a pre-test/post-test structure and a control group. The results highlighted the potential of E-learning interventions as an effective tool for promoting health behaviors, such as physical activity, in pregnant women, especially in contexts where cultural norms or access to traditional health education are limited [26].

### 3.4. The Role of Men as a Barrier or Facilitator

Although the role of social support (i.e., family members, friends, and other pregnant women) has been mentioned, the importance of the spouse’s attendance during these periods is crucial. In Iranian culture, men view themselves as more involved in significant life decisions (paternal role) and often perceive helping a spouse as a sign of weakness and a diminishment of their status [32,47,58]. The existence of gender roles in society and the traditional patriarchal culture dominant in Iranian families affect both the supportive role of their wives and contribute to negative attitudes toward men’s participation in pregnancy, delivery, and postpartum care. Participants mentioned that men’s fear of social stigma associated with participating in pregnancy and delivery care is a barrier to their involvement during this period [53,58].

Women’s preference to maintain their privacy was another reason for the limited role of men. Additionally, educational poverty affects the knowledge of men and families in prenatal care. Lack of awareness, men’s inadequate experiences, communication problems, structural problems in health centers, issues related to human resources, policymaking and managerial problems, socioeconomic barriers, and men’s occupational problems are other barriers to the role of men in supporting their wives’ pregnancy and exercise [27,53,57,58]. The results of a study demonstrated that the emotional attention and presence of men were important for their wives [21], but some were deprived of this attention and were only accompanied to clinics [32,37].

In contrast to these findings, some women commented on the benefits of PA with their partners, which helped strengthen the relationships and bonds between them [21,23]. A study assessed the effect of teaching an educational package to spouses using two methods—in-person and distance education—in childbirth preparation classes on women’s mental health [55]. The results of this study showed that the distance education method led to a decrease in physical symptoms, anxiety and insomnia, social dysfunction, and depression [55]. A majority of men preferred to receive training in pregnancy, childbirth, and postpartum care face to face, with the presence of their wives in the morning or on holidays at health centers preferably by a female doctor. In all public and private centers in Iran, maternity and postnatal services are provided by women (midwives, nurses, health personnel, and gynecologists). The results of a study by Nasiri et al. indicated that 30% of men agreed with the content of exercise during pregnancy and after childbirth [58]. Moreover, it seems that reducing the costs of pregnancy and delivery care and para-clinical measures during pregnancy, financial support from the government for expecting families, and effective implementation of the leave law would provide more relief for men and create opportunities for their participation in perinatal care [53].

### 3.5. What Strategies Increase Physical Activity Levels in Pregnancy and Postpartum?

In the following table, the strategies identified in the studies are listed (see Table 4).

## 4. Discussion

The gestational and postpartum periods provide opportunities to promote maternal health behaviors and enhance quality of life. Exercise is recognized as a positive health behavior with multidimensional benefits [4,40]. Despite guidelines recommending that women without contraindications engage in PA [1,2], fewer than 19% of women meet recommended levels of PA during their pregnancy [17] and postpartum [4]. Pregnancy and the postpartum period can affect the amount of PA women engage in due to various barriers that make maintaining adequate PA challenging. Therefore, it is crucial to explore and identify these barriers within this population [11,18,38]. In this review article, the authors addressed several questions, including the current state of PA and the identification of barriers and enablers of PA among the Iranian population. We present the following key points from the scoping review.

*First*, our review indicates that various factors, including cultural expectations, traditional beliefs, ethnicities [20,21,22,23,24,25,26,38], personal circumstances [34,43], and societal pressures limit women’s participation in PA during pregnancy and postpartum. The finding that many pregnant women prefer to exercise indoors due to the wearing of the hijab highlights the need for more accessible indoor exercise options [46]. Additionally, the economic and environmental barriers to PA [32,47], such as the high cost of exercise classes [32,35] and inadequate facilities [40,47], must be addressed to promote physical activity among these women. Postpartum women also face the challenge of balancing PA with childcare responsibilities [11,21], leading to reduced time for exercise. The impact of low income and higher numbers of pregnancies on PA participation suggests that targeted interventions for women with a lower socioeconomic status or those in later pregnancies may be particularly beneficial [46]. Moreover, societal pressure on women to prioritize family duties, along with the perception that pregnancy is a high-risk period, further compounds these barriers, highlighting the importance of social support in overcoming these obstacles [5,27,47].

*Second*, the limited involvement of spouses in promoting physical activity is a significant barrier in Iranian communities, where cultural norms reinforce traditional gender roles and care [27,37,53,55]. In these communities, men’s participation in perinatal care is often hindered by societal expectations, where women are primarily responsible for household and childcare duties, and men are expected to provide financially. Furthermore, senior women, such as mothers-in-law, hold substantial influence over caregiving, further restricting men’s participation in PA promotion [53]. Although there is some evidence of increased male involvement in perinatal care, this change is not yet widespread, and many barriers still exist. Given the central role of spouses in women’s health and well-being, further research is necessary to explore how male support can be better integrated into physical activity promotion and what strategies could help overcome these cultural constraints.

*Third*, the effectiveness of virtual training in promoting physical activity during pregnancy and postpartum is supported by emerging evidence, particularly through E-learning initiatives. These programs not only provide flexibility but also have the potential to improve mental health outcomes, demonstrating their utility as a cost-effective solution to increase engagement in PA [20,23,53,55]. The findings underscore the importance of making physical activity more accessible through virtual platforms, which can reach a broader population, particularly in areas where physical space and financial resources are limited [32]. Furthermore, enhancing the availability of sports facilities and offering free or subsidized exercise programs can eliminate some of the barriers to participation [26,44]. Some studies highlighted that E-learning [26] and mobile app [20] interventions are highly effective at increasing physical activity levels among pregnant women, particularly when in-person classes are not feasible due to constraints such as the COVID-19 pandemic [20]. These findings emphasize the significant role of technology in supporting health behaviors and overcoming the practical limitations faced by pregnant and postpartum women.

*Fourth*, the findings suggest that healthcare providers have a significant influence on physical activity levels during pregnancy and postpartum [4,27,44,49]. This influence is particularly important given the challenge of limited awareness among women about available exercise opportunities. Physicians, midwives, and other healthcare providers are in a unique position to inform and educate women on the benefits of physical activity and to guide them toward safe, accessible, and effective exercise programs [44,47]. Given that many women rely on healthcare professionals for advice during this critical period, their proactive involvement in promoting PA could help overcome one of the major barriers identified in this review stage.

*Finally*, to effectively reduce physical inactivity, it is essential to diversify the facilitators of physical activity, recognizing that some factors such as individuals (e.g., healthcare providers and experts [51]) and situations (i.e., during and after global pandemics [20,47]) have indirect effects on PA levels. The findings suggest that several promotional strategies can be effective in educating pregnant and postpartum women. These strategies include face-to-face training, mobile applications, broadcasts, SMS communications, local newspapers, billboards, and structured exercise programs [20,25,26,29,32,33,34,37,42,45,49]. Additionally, utilizing social support from neighbors, family, and friends can enhance the impact of these educational efforts [20,25,36,41,44,61]. Societies must amplify efforts to reduce physical inactivity by educating and training professional workforces. A diverse and well-trained group of professionals represents a promising opportunity to improve PA levels.

### 4.1. Strengths and Limitations

The strength of this scoping review lies in its comprehensive search, which identified 33 publications encompassing both qualitative and quantitative research across a range of racial, cultural, and educational backgrounds, as well as distinct health needs. By synthesizing findings from diverse studies, this review provides in-depth insights from women’s narratives, which can guide the development of strategies to enhance PA and healthcare during pregnancy and postpartum. However, there are limitations to consider. The studies included participants from only certain regions of Iran, meaning the results may not be representative of all Iranian women. Additionally, the review predominantly focused on women during pregnancy, with limited information on the postpartum period. Furthermore, many qualitative studies did not directly report relevant factors, potentially introducing bias.

### 4.2. Recommendation for Practice and Research

This review was designed to inform those involved in creating lifestyle interventions, guidelines, and care models for pregnant and postpartum women in Iran. The findings highlight several key criteria, barriers, facilitators, and essential components for designing effective interventions that cater to the unique needs of this population. To address the challenges that women face during pregnancy and postpartum, it is imperative to adopt a comprehensive approach that considers the contextual and socio-cultural factors that influence PA participation. Specifically, the challenges experienced by women may differ significantly between rural and urban settings, requiring interventions to be tailored accordingly. Women in urban areas often have better access to facilities, while those in rural settings may face more significant logistical and financial barriers. Therefore, it is essential to develop culturally appropriate interventions that are adaptable to local contexts. Educational programs and interventions should emphasize the importance of physical activity while incorporating local cultural beliefs and practices regarding health and exercise. Additionally, social support from family members, particularly husbands, as well as healthcare providers, plays a critical role in encouraging PA. Interventions should therefore aim to increase social support by involving both family members and healthcare providers in the promotion of PA, making it an integrated part of the routine care process.

Furthermore, this review identifies a notable gap in the training and support provided to healthcare providers and exercise instructors. These professionals play a pivotal role in facilitating PA among pregnant and postpartum women, yet they often face challenges in effectively promoting and supporting PA. To address this gap, it is recommended that training programs for healthcare providers, including physicians and midwives, be developed to enhance their ability to support women’s physical and mental health during pregnancy and postpartum. These programs should be evidence-based and culturally tailored to improve the effectiveness of healthcare providers’ recommendations. There is also a need for further research to examine the psychosocial factors that influence women’s engagement in PA during pregnancy and postpartum, particularly factors such as self-efficacy, knowledge, and social support. Research should focus on understanding how these factors vary across different populations, especially in rural versus urban areas, and how interventions can be tailored to address the specific barriers and facilitators that exist in these settings.

Finally, the limited literature on facilitators of PA during pregnancy and postpartum underscores the need for more targeted studies. Future research should explore the unique challenges and opportunities in rural and urban populations, considering cultural and regional differences. It should also investigate the role of family members, particularly husbands, in supporting PA during these periods. By addressing these gaps, future studies will contribute to a more nuanced understanding of how to design effective lifestyle interventions that meet the needs of Iranian women during pregnancy and postpartum. Ultimately, these recommendations aim to provide a clear framework for improving the design and implementation of interventions that encourage PA in this population, ensuring they are both effective and culturally appropriate.

## 5. Conclusions

Women during pregnancy and postpartum experience barriers to engaging in physical activities. The main PA barriers are captured by three themes: intrapersonal variables (physical and emotional dimensions), interpersonal factors (cultural poverty, social beliefs, and the role of family members), and the environment (organizational structures and economic situation). This review article discusses how pregnancy and the postpartum period provide an excellent opportunity to educate women about the benefits of PA and a healthy lifestyle. Raising women’s awareness of PA during pregnancy and postpartum, as well as changing their attitudes, may lead to behavioral adjustments. Meanwhile, the role of clinicians is very prominent. Healthcare providers encounter additional barriers while offering care to pregnant and postpartum women, such as skill limitations and a lack of communication between practical interventions and clinical concerns. Understanding the views of two key stakeholders—women and healthcare providers—is critical for developing effective interventions. This review highlights some facilitators and barriers that are crucial to the design and implementation of PA interventions for pregnant and postpartum Iranian women (see Figure 2).

## Figures and Tables

**Figure 1 healthcare-12-02416-f001:**
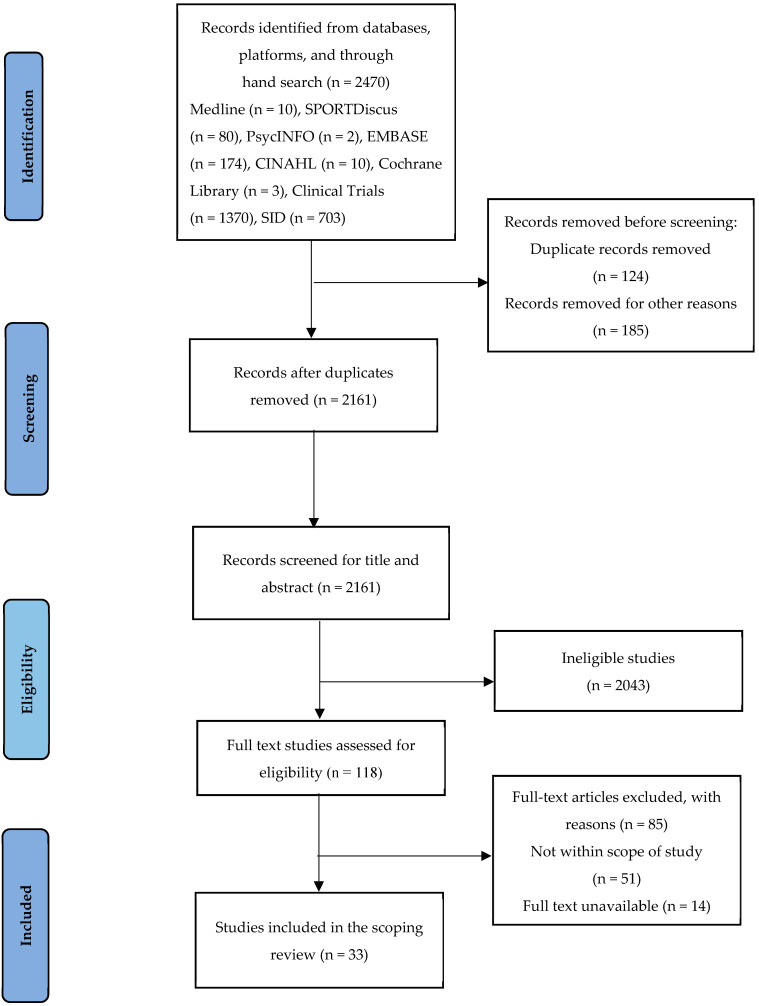
Study selection PRISMA flowchart.

**Figure 2 healthcare-12-02416-f002:**
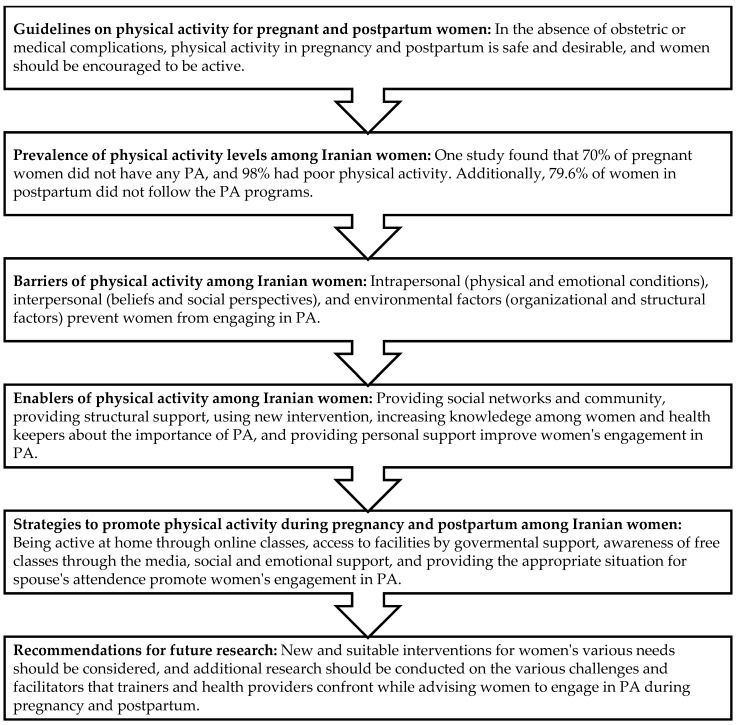
Infographic of the scoping review. This figure illustrates the main findings related to physical activity during pregnancy and postpartum in Iranian women. Guidelines for physical activity, prevalence levels, and the barriers and enablers to physical activity are shown, with specific recommendations for promoting physical activity. References such as Hajimiri et al. [11], Arefi et al. [36] and Dolatabadi et al. [48] highlight significant barriers, while other sources [19,29,32,51] discuss effective interventions for improving engagement in physical activity. Further, strategies to promote PA are discussed through social and structural supports [20,26,32,53].

**Table 1 healthcare-12-02416-t001:** Inclusion and exclusion search criteria for electronic database search.

Inclusion Criteria	Exclusion Criteria
Iranian-based research articles	Abstracts without full text
Iranian women in pregnancy and postpartum periods	Articles focusing on the participant, not within the scope of this review (e.g., investigating the barriers to and facilitators of PA in women who are not in these periods)
Articles published in peer-reviewed or grey literature	Articles that did not include the concepts of facilitators of and barriers to PA for this population
Articles published in Persian and English	Articles not in Persian or English language
Qualitative and quantitative research designs including, but not limited to, natural experiments with pre–post measures, content analysis, systematic or non-systematic reviews, commentary, guidelines and manuals, policy, or practice papers	Non-Iranian-based research articles
Articles stating the context and concept of barriers and facilitators of PA levels	Duplicated articles in Persian and English

**Table 2 healthcare-12-02416-t002:** Main characteristics of studies.

Title	Author(s), Year	Study Design	Description/Objective(s)	Participants	Key Findings
“Preparation of Protocol for Removing Physical, Psychological, Environmental, and Social Barriers to Physical Activity in Pregnant Women”	Mokhtari et al. [32], 2019	Descriptive analytics method (cross-sectional studies)	To develop a protocol to remove barriers of physical activity in healthy pregnant women admitted to Shahid Beheshti Health Center and Hospital in Isfahan	380 pregnant women and experts	To promote and increase the amount of physical activity during pregnancy, attention and recommendation to implement this protocol in routine prenatal care is essential. The authors assessed 61 solutions in physical, mental, social, and environmental dimensions and introduced as the facilitators of PA.
“Examining Exercise Behavior Beliefs of Pregnant Women in Second and Third Trimester: Using Health Belief Model”	Moridi et al. [33], 2016	Pre–post test (intervention)	To examine women’s exercise behavior from their second to third pregnancy trimester using the health belief model (HBM)	100 pregnant women	The educational program based on the HBM could provide pregnant women with a conceptual framework to improve their beliefs regarding exercise.
“The Role of Perceived Barriers in Postpartum Women’s Health-Promoting Lifestyle: A Partial Mediator between Self-Efficacy and Health-Promoting Lifestyle”	Hajimiri et al. [11], 2017	Cross-sectional study with mediational analysis	To determine the effects of self-efficacy and perceived barriers on Iranian women’s health-promoting lifestyle (HPL) in the first year after childbirth	310 women in postpartum	Self-efficacy not only promotes women’s HPL but also indirectly affects women’s lifestyles by reducing perceived barriers.
“Effect of Educational Package on the Lifestyle of Primiparous Mothers During the Postpartum Period: A Randomized Controlled Clinical Trial”	Khodabandeh et al. [34], 2017	Randomized clinical trial (intervention)	To determine the effects of a lifestyle educational package in primiparous women	220 primiparous women	The training provided positively affected certain health behaviors in the mothers; however, it failed to improve their physical activity level and nutritional status.
“Monitored Home-Based with or without Face-to-Face Exercise for Maternal Mental Health during the COVID-19 Pandemic”	Veisy et al. [35], 2022	Randomized controlled trial (RCT) with three parallel arms	To investigate adherence to face-to-face plus monitored home exercise versus monitored home-based exercise alone on maternal mental health and outcomes	150 women during pregnancy and postpartum	It is not reported yet.
“Development and Psychometric Properties of the Physical Activity Scale for Pregnant Women”	Arefi et al. [36], 2022	Sequential explanatory mixed-methods design (qualitative and quantitative methods)	To develop a scale for measuring physical activity based on social cognitive theory in pregnant Iranian women	240 pregnant women	The psychometric properties of the physical activity scale are valid and reliable scales that can help us better understand aspects associated with physical activity in pregnant women.
“Association between Perceived Social Support and Health-Promoting Lifestyle in Pregnant Women: A Cross-Sectional Study”	Fathnezhad et al. [27], 2021	Quantitative cross-sectional study	To investigate the association between social support and health-promoting lifestyle in pregnancy	360 pregnant women	Pregnant women with better perception of social support had a better performance in adopting health-promoting lifestyle changes like PA.
“Self-Care Education Needs in Gestational Diabetes Tailored to the Iranian Culture: A Qualitative Content Analysis”	Kolivand et al. [37], 2018	Qualitative research	To determine the needs of women as an essential first step to formulating a self-care guide fitting the Iranian culture	13 diabetic pregnant women and 10 care providers	There are main aspects of self-care educational/supportive needs in domains of lifestyle, awareness and capability, mental health, and family that can affect the physical activity level.
“Development and Psychometric Testing of the ‘Barriers to Physical Activity during Pregnancy Scale’ (BPAPS)”	Amiri Farahani et al. [38], 2021	Mixed methods (qualitative and quantitative methods)	To develop and validate a scale to assess barriers to physical activity in a pregnant population	320 pregnant women and 10 experts	The final BPAPS could realize the barriers of PA during pregnancy and include four factors, including pregnancy-related intrapersonal barriers, non-pregnancy-related intrapersonal barriers, interpersonal barriers, and environmental barriers.
“The Study of Knowledge, Attitude, and Practice of Puerperal Women about Exercise during Pregnancy”	Noohi et al. [24], 2010	Correlation study design (cross-sectional studies)	To determine the knowledge, attitude, and practice of puerperal women admitted to Kerman hospitals about exercise during pregnancy	256 pregnant women	Mothers’ concern about exercise during pregnancy is due to their knowledge deficit about permitted exercises during pregnancy. Giving information and education would be helpful to promote their practice.
“Evaluation of Women’s Exercise and Physical Activity Beliefs and Behaviors during Their Pregnancy and PostPartum Based on the Planned Behavior Theory”	Garshasbi et al. [21], 2020	Descriptive analytics method (cross-sectional studies)	To evaluate beliefs and physical performance during pregnancy and postpartum	200 women within 1 year of a child’s birth	Compared to during pregnancy, physical activity decreased after delivery. Despite pregnancy and postpartum promoting a sedentary lifestyle, this period is an ideal time to engage with healthcare centers to educate and encourage women to change their attitudes, beliefs, and behaviors towards physical activity.
“Assessing Physical Activity Self-Efficacy and Knowledge about Benefits and Safety during Pregnancy among Women”	Akbari et al. [39], 2016	Descriptive analytics method (cross-sectional studies)	To assess physical activity level, self-efficacy, and knowledge about benefits and safety during pregnancy	205 pregnant women	Education played an important role in women’s information about health benefits and safety knowledge related to physical activity during pregnancy.
“The Effect of the 5A Model on Behavior Change of Physical Activity in Overweight Pregnant Women”	Ghaderpanah et al. [29], 2017	Randomized clinical trial (intervention)	To determine the effect of the 5A model on behavior change of physical activity in overweight pregnant women	120 overweight pregnant women	The self-management intervention based on the 5A model has an effective role in changing pregnant women’s behavior and can lead to motivation, attitude, and behavioral modifications.
“Trend of Exercise before, during, and after Pregnancy”	Esmaelzadeh et al. [4], 2008	Descriptive analytics method (cross-sectional studies)	To determine the trend of mother’s physical exercise before, during, and after pregnancy and postpartum	225 postpartum women	The trend of physical activity regressed from the period before pregnancy to 3 months after. The authors suggested that professional care try to advance exercise among women during pregnancy and the postpartum period.
“Application of BASNEF Model to Predict Postpartum Physical Activity in Mothers Visiting Health Centers in Kermanshah”	Ouji et al. [40], 2014	Descriptive analytics method (cross-sectional studies)	To identify predictors of postpartum physical activity among women visiting health centers in Kermanshah	400 postpartum women	The majority of mothers were inactive during the postpartum period. Relevant interventions should be designed to modify mothers’ behavioral intentions and promote physical activity after childbirth.
“The Effect of Educational Intervention on Physical Activity Self-Efficacy and Knowledge about Benefits and Safety among Pregnant Women”	Mousavi et al. [41], 2020	Pre–post test (intervention)	To determine the effect of an educational intervention to improve self-efficacy and knowledge of physical activity benefits and safety among pregnant women	144 pregnant women referred to health centers in south Tehran	Conducting educational interventions can be effective to improve pregnant women’s awareness about the advantages and benefits of physical activity and to improve their self-efficacy during pregnancy.
“The Effect of Trans Theoretical Model (TTM) on Exercise Behavior in Pregnant Women Referred to Dehaghan Rural Health Center”	Solhi et al. [42], 2012	Quasi-experimental study, pre–post test	To assess the effect of the Trans Theoretical Model (TTM) on physical activity in pregnant women	100 pregnant women	The application of TTM for physical activity intervention indicates that the procedure was very effective in improving attitudes and encouraging women to exercise more during pregnancy.
“Perceptual Factors (Awareness, Attitude) and Positive, Indeterminate, and Negative Coping Strategies in Pregnant Women on Regular Physical Activity”	Kianfard et al. [43], 2021	Quasi-experimental study, pre–post test	To determine the effect of perceptual factors (awareness, attitude) and positive, intermediate, and negative nurturing factors on the physical activity of pregnant women visiting health centers in Tehran	250 pregnant women	E-learning can increase the PA level. Also, the administration of pregnancy training classes promotes behaviors related to a healthy lifestyle and leads to increased physical activity.
“Facilitators, Barriers, and Structural Determinants of Physical Activity in Nulliparous Pregnant Women: A Qualitative Study”	Kianfard et al. [44], 2022	Qualitative methods (cross-sectional study)	To recognize the facilitators, barriers, and structural factors that influence activity among pregnant women	52 pregnant women	Lack of awareness and misinformation, accessibility obstacles, and economic problems are the worst physical activity barriers during pregnancy. Being among other pregnant women and the physicians’ recommendations are the best facilitators of physical activity during pregnancy.
“The Effect of Education Based on Self-Efficacy Strategies in Changing Postpartum Physical Activity”	Abdollahi et al. [45], 2016	Randomized controlled trial (intervention)	To examine self-efficacy as a moderator on changes in postpartum physical activity	80 postpartum women	According to the effect of self-efficacy on physical activity, it is suggested that workshops should be held in health centers based on self-efficacy physical activity strategies for postpartum women.
“Knowledge and Performance of Pregnant Women Referring to Shabihkhani Hospital on Exercises During Pregnancy and PostPartum Periods”	Abedzadeh et al. [17], 2011	Descriptive statistical method (cross-sectional study)	To review the knowledge and the performance level of pregnant women about exercise during pregnancy and postpartum in pregnant women	200 pregnant women	Most of the women had moderate knowledge about physical activities and performed poorly. Therefore, the authors emphasized educating women about the importance of exercise and its role in improving their performance during pregnancy.
“Exploring the Intensity, Barriers and Correlates of Physical Activity in Iranian Pregnant Women: A Cross-Sectional Study”	Ahmadi et al. [46], 2021	Correlation study design (cross-sectional study)	To determine the intensity, barriers to, and correlates of physical activity (PA) in pregnant Iranian women	300 pregnant women	Encouraging individuals to be more physically active before pregnancy and enhanced support from family and spouses who incite women to exercise during and after pregnancy may increase the intensity of PA most effectively.
“Weight Management Challenges in Nulliparous Women Being Overweight or Obese Due to Pregnancy: A Qualitative Study”	Heidari Dehui et al. [47], 2023	Qualitative research (conventional qualitative content analysis method)	To explain the challenges in performing post-pregnancy weight-management behaviors in nulliparous women who are overweight and obese due to pregnancy	15 pregnant women	Motivational and support programs should be developed for all involved people, including the mother herself, her family, and health personnel. Improving healthy behaviors not only requires relevant stakeholders’ commitment but also demands women, their families, and communities’ intention to engage in healthy behaviors.
“Barriers to Physical Activity in Pregnant Women Living in Iran and Its Predictors: A Cross-Sectional Study”	Dolatabadi et al. [48], 2022	Correlation study design (cross-sectional study)	To determine the barriers of PA and its predictors in pregnant Iranian women	300 pregnant women	Raised awareness and education about the benefits of PA during pregnancy and more support from family and spouses may increase PA during pregnancy. PA interventions including increased access to sports facilities, and gymnasiums need to be targeted toward those with lower levels of education and income whose PA levels are low.
“Determine of Facilitators, Barriers, and Structural Factors of Physical Activity in Nulliparous Pregnant Women: A Qualitative Study Using Maxqda”	Kianfard et al. [26], 2022	Qualitative study (interview and content analysis)	To recognize facilitators of, barriers to, and structural influences on behavior of physical activityamong pregnant women	30 pregnant women	The need for sufficient information on the advantages of physical activity and the role of nurturing factors were the important obstacles to PA. Also, facilitating factors included using E-learning to educate pregnant women and awareness of the existence of sports classes.
“Investigating the Effect of Positive, Intermediate, and Negative Enabling and Training Factors Affecting Physical Activity in Pregnant Women”	Kianfard et al. [23], 2021	Randomized controlled trial (pre–post test, intervention)	To investigate the implementation of comprehensive educational methods based on the conditions and facilities of the educational environment	250 pregnant women	Intervention based on E-learning has a significant effect on training and enabling factors to increase physical activity in pregnant women.
“Mothers’ Views on Mobile Health in Self-Care for Pregnancy: A Step towards Mobile Application Development”	Pouriayevali et al. [49], 2023	Qualitative study (interviews and conventional content analysis)	To explore mothers’ views on mobile health in self-care for pregnancy to develop a mobile application	14 pregnant women	The findings showed the amount, intensity, and type of PA that women need in pregnancy. Also, findings can be useful for designing an APSC that is localized based on the operational needs of pregnant women to make them capable and self-caring in controlling pre-risk situations.
“Kegel Exercise Application during Pregnancy and PostPartum in Women Visited at Hamadan Health Care Centers”	Riyazi et al. [25], 2007	Qualitative study (interviews and conventional content analysis)	To determine the rate of making use of Kegel exercise during pregnancy and postpartum	245 women in pregnancy and postpartum	Women were not familiar with the Kegel exercise and one of the reasons may be insufficient attention of healthcare providers to the Kegel exercise.
“Effectiveness of a Group-Based Educational Program on Physical Activity among Pregnant Women”	Shakeri et al. [50], 2012	Randomized controlled trial (pre–posttest, intervention)	To assess the effectiveness of a group educational program on pregnant women’s physical activity	280 pregnant women	The group-based educational program seemed to promote physical activity during pregnancy. These kinds of programs should be implemented in prenatal clinics.
“The Impact of Education on Performing Postpartum Exercise Based on Health Belief Model”	Safarzadeh et al. [22], 2014	Randomized controlled trial (intervention)	To evaluate the impact of education, based on the health belief model, on performing postpartum exercise in primipara Iranian women	195 women in postpartum	Health belief model-based education has positive impacts on behavioral improvement and PA level.
“Survey of Midwives’ Practice and Its Related Factors toward Exercise Instruction during Pregnancy in Health Centers of Isfahan in 2011”	Shayanmanesh et al. [51], 2013	Descriptive statistical method (correlation study design)	To determine midwives’ practice and its related factors toward exercise training during pregnancy	40 health care providers and midwives	All participants had weak performance, and most midwives had intermediate knowledge of exercise training during pregnancy.
“Examining How to Exercise and the Effective Factors on it Based on the Perspective of Pregnant Women Referring to Health Centers in Astara City”	Kiani et al. [52], 2012	Descriptive study (cross-sectional study)	To investigate the reasons for the tendency of pregnant women in Astara to exercise during pregnancy	200 pregnant women	The findings highlighted the significant role of doctors, midwives, and healthcare personnel in influencing women’s willingness to exercise during pregnancy. Additionally, the attitudes of these professionals towards exercising during pregnancy should be evaluated and corrected as needed.
“Mobile-Application Intervention on Physical Activity of Pregnant Women in Iran during the COVID-19 Epidemic in 2020”	Kiani et al. [20], 2021	Quasi-experimental design (intervention)	To determine the impact of an educational intervention based on mobile apps for physical activity in pregnant women	93 pregnant women	Mobile apps can be used to promote physical activity in pregnant women. Therefore, it is recommended that mobile app education should be applied to face-to-face classes in health centers for physical activity in pregnant women during the pandemic situation.

**Table 3 healthcare-12-02416-t003:** Themes, sub-themes, and categories and sub-categories of barriers and facilitators of PA during pregnancy and postpartum.

Theme	Subtheme	Categories	Subcategories
Barriers of PA	1. Intrapersonal Factors	Health-related barriers	Physical limitations (e.g., tiredness, dizziness, fatigue) Health conditions (e.g., gestational diabetes, high blood pressure) Physiological changes during pregnancy (e.g., abdominal growth, shortness of breath)
		Non-health-related barriers	Lack of motivation Low self-efficacy Lack of knowledge about exercise programs Lack of awareness of pregnancy/postpartum-specific PA guidelines
	2. Interpersonal Factors	Family and social support	Family discouragement (e.g., husbands not allowing exercise) Role of spouses and family in discouraging PA Lack of communication between healthcare providers and patients
		Cultural and religious beliefs	Cultural restrictions on women’s physical activity (e.g., modesty in dress) Negative perceptions of women involved in exercise Religious expectations regarding women’s lifestyle and appearance
	3. Environmental Factors	Physical environment:	Lack of access to suitable facilities (e.g., gyms, outdoor spaces) Safety concerns in the environment (e.g., air pollution, crowded spaces) Limited space at home for physical activity
		Socioeconomic factors:	Financial constraints (e.g., high cost of exercise classes, limited financial resources) Lack of information on free exercise programs
		Work-related constraints	Long hours of sitting due to desk-based work Lack of time and energy to exercise due to caregiving responsibilities
Facilitators of PA	1. Social Support and Community Networks	Family and spousal support	Encouragement and participation of husbands or family members (e.g., accompanying during exercise) Emotional support from friends and social networks
		Community support	Neighborhoods that are safe for walking or outdoor exercise Community-based activities and facilities for exercise Role of social networks in motivating regular physical activity
	2. Educational and Intervention Programs	Health education and awareness	Providing information about the benefits and safety of physical activity E-learning and online platforms for exercise programs Counseling through various methods (e.g., phone, SMS, face-to-face)
		Exercise intervention methods	Online and home-based exercise programs Structured face-to-face exercise programs Educational materials (e.g., booklets, videos) on safe physical activities during pregnancy/postpartum
		Training for healthcare providers	Educating healthcare professionals to support pregnant and postpartum women in engaging in physical activity
	3. Structural Support	Access to facilities	Government or workplace initiatives to provide facilities or financial support for exercise classes Availability of postpartum fitness programs or rehabilitation
		Policy and program support	Incentives for exercising (e.g., bonuses, rewards for attending programs) Financial assistance for exercise classes or personal trainers
	4. Personal Factors	Increased self-efficacy	Confidence in the ability to engage in physical activity Encouragement to start exercise programs
		Motivation and attitude change	Shifting attitudes towards physical activity (from passive to active participation) Women’s personal desire to improve health or maintain fitness during pregnancy/postpartum

**Table 4 healthcare-12-02416-t004:** Evidence from Iranian studies and strategies for promoting physical activity.

Author(s)	Title	Results
Shakeri et al. [50]	“Effectiveness of a Group-Based Educational Program on Physical Activity among Pregnant Women”	The study assessed the effectiveness of a group educational program on pregnant women’s physical activity. Women in the experimental group attended eight 90-min sessions covering anatomical and physiological changes, the effects of physical activity during pregnancy, and Q&A counseling and group exercises. Questionnaire results indicated that the program promoted physical activity during pregnancy. The study suggested implementing such programs in prenatal clinics.
Moridi et al. [33]	“Examining Exercise Behavior Beliefs of Pregnant Women in Second and Third Trimester: Using the Health Belief Model”	The intervention involved eight 40-min exercise classes with 15-day intervals. Sessions included neuromuscular training, patterned breathing, proper positions during pregnancy and labor, and 30 min of practical exercise. The health belief model-based educational program improved perceived threats, benefits, barriers, cues to action, and self-efficacy in participants, enhancing their beliefs about exercise during pregnancy.
Hajimiri et al. [11]	“The Role of Perceived Barriers in the PostPartum Women’s Health-Promoting Lifestyle: A Partial Mediator between Self-Efficacy and Health-Promoting Lifestyle”	Self-efficacy was measured using the Self-Rated Abilities for Health Practices Questionnaire, assessing nutrition, exercise, psychological well-being, and responsible health practices. Perceived barriers, measured by the Barriers to Health-Promoting Activities for Disabled Persons Scale, were the mediator variable. Health-promoting lifestyle (HPL), the dependent variable, was measured using six dimensions: physical activity, health responsibility, nutrition, stress management, spiritual growth, and interpersonal relations. The study found that low self-efficacy predicted physical activity levels, and improving self-efficacy could address barriers such as postpartum depression, loss of quality of life, lack of social support, breastfeeding, and baby care among postpartum women.
Khodabandeh et al. [34]	“Effect of Educational Package on Lifestyle of Primiparous Mothers during Postpartum Period: A Randomized Controlled Clinical Trial”	In the intervention group, mothers received face-to-face, phone, and SMS counseling as well as a booklet in addition to routine postpartum training, covering physical activity, diet, mental health, and postpartum health problem prevention. Data were collected using the Health Behaviors, Food Frequency, and International Physical Activity Questionnaires. The findings showed that the training positively impacted certain health behaviors in mothers but did not improve their physical activity level and nutritional status.
Veisy et al. [35]	“Monitored Home-Based with or without Face-to-Face Exercise for Maternal Mental Health during the COVID-19 Pandemic”	In two groups (face-to-face plus monitored home exercise vs. only monitored home-based exercise), the exercise programs were performed and participants were assessed at specific intervals during pregnancy and postpartum. The exercise diary was used to assess adherence, while the Edinburgh Depression Scale and the Positive and Negative Affect Schedule assessed prenatal and postnatal depression and affect, respectively. This study stated that the cost of current exercise methods is an important barrier for women during pregnancy and postpartum. Thus, combining face-to-face with monitored home exercise could effectively improve women’s health at a lower cost compared to conventional supervised exercise interventions.
Arefi et al. [36]	“Development and Psychometric Properties of the Physical Activity Scale for Pregnant Women”	This study aimed to identify the factors affecting the physical activity (PA) of pregnant women based on the framework of social cognitive theory and to develop a standard and practical tool for further research. Conducted in two stages, the initial phase involved designing and validating items in Persian. A qualitative study identified key elements of PA, and an exploratory factor analysis (EFA) was performed to determine the major factor structure, followed by confirmatory factor analysis (CFA) for coherence. The findings indicated that self-efficacy had the highest effect on PA in pregnant women, revealing a six-factor structure: self-efficacy, self-regulation, family support, friend support, outcome expectancy, and self-efficacy in overcoming barriers.
Fathnezhad et al. [27]	“Association between Perceived Social Support and Health-Promoting Lifestyle in Pregnant Women: A Cross-Sectional Study”	The goal of this study was to investigate the association between social support and health-promoting lifestyle during pregnancy. Data were collected using three questionnaires: a self-reported demographic and obstetric questionnaire, the Health-Promoting Lifestyle Profile, and the Perceived Social Support Questionnaire. The findings showed that the dimensions of spiritual growth, nutrition, health responsibility, stress management, and physical activity had scores ranging from highest to lowest. According to the results, barriers to adopting a healthy lifestyle included a lack of support from healthcare providers and family, family issues (especially with the spouse), and negative interactions with healthcare providers.
Kolivand et al. [37]	“Self-Care Education Needs in Gestational Diabetes Tailored to the Iranian Culture: A Qualitative Content Analysis”	This study aimed to identify the self-care needs of women to develop a guide that fits Iranian culture. Semi-structured interviews, each lasting no more than 50 min, were conducted. The findings highlighted the main aspects of self-care educational/supportive needs, particularly in lifestyle (physical activity and diet), awareness and capability, mental health, and family. The results emphasized the importance of these needs in creating comprehensive self-care educational programs, focusing on physical activity, mental health, family roles, and religious interests.
Amiri Farahani et al. [38]	“Development and Psychometric Testing of the ‘Barriers to Physical Activity during Pregnancy Scale’ (BPAPS)”	Pregnancy can affect the amount of physical activity that women engage in, making it challenging to ensure adequate activity levels. This study aimed to explore and identify barriers to physical activity in pregnant women through a two-phase approach. Phase 1 involved a comprehensive literature review and extraction of scale items, while Phase 2 focused on determining the psychometric properties of these items. Qualitative and quantitative face validity were assessed through interviews with pregnant women. The study developed and validated the BPAPS, a scale with 29 items across four factors: pregnancy-related intrapersonal barriers, non-pregnancy-related barriers, interpersonal barriers, and environmental barriers.
Noohi et al. [24]	“The Study of Knowledge, Attitude, and Practice of Puerperal Women about Exercise during Pregnancy”	This study aimed to assess the knowledge, attitude, and practice of puerperal women in Kerman hospitals regarding exercise during pregnancy. A researcher-made questionnaire based on the theory of planned behavior was used, covering demographic characteristics, knowledge, attitude, and practice regarding exercise. Face-to-face interviews were conducted for data collection. The results showed a weak positive correlation between knowledge, attitude, and practice. The findings suggest that maternal concerns about exercise stem from a lack of knowledge about safe and beneficial exercises. Providing information and education could improve mothers’ knowledge, attitudes, and practices regarding exercise during pregnancy.
Garshasbi et al. [21]	“Evaluation of Women’s Exercise and Physical Activity Beliefs and Behaviors during Their Pregnancy and Postpartum Based on the Planned Behavior Theory”	The study aimed to evaluate beliefs and physical performance during pregnancy and postpartum. The Global Physical Activity Questionnaire and the Exercise Beliefs Questionnaire were used to assess physical activity and exercise beliefs (behavioral, normative, and control). Key behavioral beliefs included improvements in mood and fitness, while common normative beliefs involved influences from spouses, mothers, and healthcare employees. Control beliefs such as fatigue, lack of energy, and time constraints were prevalent. The study concluded that physical activity generally decreases after delivery, and both pregnancy and postpartum periods tend to encourage a sedentary lifestyle.
Akbari et al. [39]	“Assessing Physical Activity Self-Efficacy and Knowledge about Benefits and Safety during Pregnancy among Women”	The study aimed to assess physical activity levels, self-efficacy, and knowledge about the benefits and safety of exercise during pregnancy. Using a self-reported questionnaire, IPAQ, and a demographics form, the study found a significant relationship between physical activity self-efficacy and maternal education and activity levels. No significant links were found between self-efficacy, age, and gestational status. Only 33.1% of participants were aware that exercise could boost maternal energy. Thus, improving education on the benefits and safety of physical activity during pregnancy is essential.
Ghaderpanah et al. [29]	“The Effect of 5A Model on Behavior Change of Physical Activity in Overweight Pregnant Women”	The study aimed to assess the impact of the 5A model on physical activity behavior in overweight pregnant women. Participants in the intervention group received face-to-face training based on the 5A self-management model over 3 months, following five steps: question, assessment, guidance, agreement, help, and follow-up. The group sessions aimed to increase PA, with participants tracking their weekly activities. The study found that the 5A model significantly enhanced behavior change, motivation, and attitudes toward physical activity and helped manage weight gain in line with guidelines.
Esmaelzadeh et al. [4]	“Trend of Exercise before, during, and after Pregnancy”	The study aimed to assess physical activity (PA) levels before, during, and after pregnancy. Data were collected through a questionnaire with three sections: demographics, exercise activity, and exercise beliefs. The findings indicated a decrease in PA levels from before pregnancy to 3 months postpartum, suggesting that professional care providers should focus on encouraging continued PA in prenatal and postpartum settings.
Ouji et al. [40]	“Application of BASNEF Model to Predict Postpartum Physical Activity in Mothers Visiting Health Centers in Kermanshah”	The study used the Beliefs, Attitudes, Subjective Norms, and Enabling Factors (BASNEF) model to identify predictors of postpartum physical activity among women in Kermanshah, Iran. Questionnaires measured demographic variables, BASNEF constructs, and physical activity levels. Results showed that 83% of mothers had low physical activity. Knowledge, attitude, and subjective norms significantly predicted the intention to engage in PA, with behavioral intention being the strongest predictor. Interventions should focus on altering behavioral intentions to increase postpartum physical activity.
Mousavi et al. [41]	“The Effect of Educational Intervention on Physical Activity Self-Efficacy and Knowledge about Benefits and Safety among Pregnant Women”	The study aimed to evaluate the effectiveness of an educational intervention in improving self-efficacy, knowledge about the benefits of PA, and safety tips for pregnant women. Data were collected using several questionnaires. The intervention involved face-to-face or small group sessions where participants received comprehensive information about PA during pregnancy, including its benefits, safety guidelines, and strategies to enhance self-efficacy. Additionally, participants attended an in-person training session at the center three weeks later to cover further content and address any questions. Each face-to-face session lasted between 45 to 60 min, and participants had the opportunity to ask questions at the end of each session. Following the intervention, there was a significant increase in PA self-efficacy and an improvement in knowledge regarding the benefits and safety of physical activity. The study concluded that such educational interventions are effective in enhancing pregnant women’s awareness and self-efficacy concerning physical activity.
Solhi et al. [42]	“The Effect of Trans Theoretical Model (TTM) on Exercise Behavior in Pregnant Women Referred to Dehaghan Rural Health Center”	Despite the known benefits of PA during pregnancy, many women do not engage in it, likely due to a lack of knowledge. The study aimed to evaluate the effectiveness of the Transtheoretical Model (TTM) in promoting physical activity among pregnant women. The intervention consisted of five one-hour sessions that included discussions, lectures, and films about PA, its benefits, and perceived barriers. Data were collected using a researcher-developed questionnaire based on the standard questionnaire for physical activities during pregnancy and TTM. The results support the effectiveness of using TTM for PA interventions. The study found that the intervention significantly improved attitudes toward exercise and encouraged greater physical activity among pregnant women.
Kianfard et al. [43]	“Perceptual Factors (Awareness, Attitude) and Positive, Indeterminate, and Negative Nurturing Factors Affecting Physical Activity of Pregnant Women Visiting Healthcare Centers in Tehran: Examination and Analyses”	The study aimed to investigate the impact of perceptual factors (awareness and attitude) and nurturing factors (positive, intermediate, and negative) on physical activity among pregnant women visiting health centers in Tehran. The research was conducted in two stages. For the qualitative stage, a literature review was followed by semi-structured interviews with pregnant women to identify criteria for increasing PA. For the quantitative stage, based on the findings from the first stage, an E-learning program intervention was implemented using the PEN-3 cultural model. This quasi-experimental study assessed the effect of the E-learning program on PA. The results showed that the E-learning intervention had a statistically significant effect on perceptual factors and effectively increased PA among pregnant women.
Kianfard et al. [26]	“Determine Facilitators, Barriers, and Structural Factors of Physical Activity in Nulliparous Pregnant Women: A Qualitative Study Using Maxqda”	The study aimed to identify facilitators, barriers, and structural factors affecting PA among nulliparous pregnant women. Using Maxqda for qualitative analysis, several themes were extracted from open-ended responses regarding obstacles to PA during pregnancy. The facilitators were nurturing factors such as socio-cultural, socioeconomic, and individual elements. The barriers were socio-cultural and socioeconomic challenges as well as individual and structural issues, including environmental and organizational constraints. The findings highlight the need to address structural problems in the community to enhance facilities and support for pregnant women to engage in PA during pregnancy.
Kiani et al. [20]	“Mobile-Application Intervention on Physical Activity of Pregnant Women in Iran during the COVID-19 Epidemic in 2020”	This study assessed the impact of a mobile app-based educational intervention on physical activity in pregnant women during the COVID-19 pandemic. Utilizing validated questionnaires, the intervention covered 12 domains, including a description of PA, various types of proper pregnancy exercises, exercise planning, massage techniques, stretching, relaxation techniques, reminders for important exercise points, exercise demonstration movements, and educational videos. The results indicated that the mobile app significantly promoted physical activity, with high scores for perceived benefits and enjoyment. The study suggested that mobile app education, combined with face-to-face classes, can effectively enhance PA among pregnant women, particularly in a pandemic context.
Kianfard et al. [23]	“Investigating the Effect of Positive, Intermediate and Negative Enabling and Training Factors Affecting Physical Activity in Pregnant Women”	The goal of this qualitative study was to identify new approaches to the underlying factors of PA in pregnant women. The study used the Pregnancy Physical Activity Questionnaire (PPAQ) and a questionnaire designed based on needs assessment results and the dimensions of the PEN-3 model. This quasi-experimental study evaluated the effect of an E-learning program on PA. The results indicated that the E-learning intervention significantly impacted enabling and training factors, effectively increasing PA in pregnant women.
Ahmadi et al. [46]	“Exploring the Intensity, Barriers and Correlates of Physical Activity in Iranian Pregnant Women: A Cross-Sectional Study”	The study aimed to determine the intensity, barriers, and correlates of PA among Iranian pregnant women using a demographic and obstetrical history questionnaire, the Pregnancy Physical Activity Questionnaire, and the Exercise Benefits/Barriers Scale. Results indicated that the highest barriers were time expenditure and family discouragement. PA intensity was significantly associated with factors such as pre-pregnancy or early pregnancy body mass index, ethnicity, education level, number of children, gestational age, participation in childbirth preparation classes, habitual exercise before pregnancy, and income. PA barriers were significantly associated with ethnicity, income, and habitual exercise before pregnancy. To enhance PA intensity, it is essential to encourage physical activity before pregnancy and provide support from friends and family during pregnancy, particularly targeting women with a low income and education.
Kianfard et al. [44]	“Facilitators, Barriers, and Structural Determinants of Physical Activity in Nulliparous Pregnant Women: A Qualitative Study”	This study aimed to identify facilitators, barriers, and structural factors influencing physical activity among pregnant women. Using open-ended questions, researchers identified six themes categorized into the PEN-3 model’s facilitators, barriers, and structural factors. Key barriers included lack of awareness, misinformation, accessibility issues, and economic problems, while the primary facilitators were social support from other pregnant women and recommendations from physicians.
Dolatabadi et al. [48]	“Barriers to Physical Activity in Pregnant Women Living in Iran and Its Predictors: A Cross-Sectional Study”	The study aimed to identify barriers to PA and its predictors in Iranian pregnant women using the Pregnancy Physical Activity Questionnaire and the Barriers to Physical Activity during Pregnancy Scale. The findings indicated that interpersonal barriers were the most prevalent. It is recommended that perinatal care providers actively encourage, educate, and reassure pregnant women, their spouses, and their families about the benefits and safe practices of PA during pregnancy.
Shayanmanesh et al. [51]	“Survey of Midwives’ Practice and Its Related Factors toward Exercise Instruction during Pregnancy in Health Centers of Isfahan in 2011”	The study aimed to assess midwives’ practices and related factors concerning exercise training during pregnancy. Data were collected through two questionnaires and an observation checklist. The questionnaires covered midwives’ characteristics, sources of information, and perceived obstacles to exercise training, as well as their knowledge and attitudes toward exercise. The observation checklist evaluated their performance in this area. The results revealed that most midwives cited a lack of training and time as barriers to providing exercise training. Many midwives had intermediate knowledge and neutral attitudes about exercise training. A significant relationship was found between practice and demographic variables, with those citing time constraints showing poorer performance. Overall, midwives demonstrated weak performance in exercise training, and most had only intermediate knowledge of the topic.
Safarzadeh et al. [22]	“The Impact of Education on Performing Postpartum Exercise Based on Health Belief Model”	The study aimed to evaluate the impact of health belief model-based education on postpartum exercise in Iranian primiparous women. The intervention included four 60-min sessions of theoretical and practical training on PA, focusing on perceived susceptibility, severity, barriers, benefits, self-efficacy, and action guidelines. A control group received a CD and training booklet. The results showed that the health belief model-based education significantly improved postpartum PA, with increased scores for perceived susceptibility, severity, benefits, barriers, action guidance, and self-efficacy.
Riyazi et al. [25]	“Kegel Exercise Application during Pregnancy and Postpartum in Women Visited at Hamadan Health Care Centers”	The study aimed to determine the prevalence of Kegel exercise use during pregnancy and postpartum. Data were collected through questionnaires and interviews, revealing that many women were unfamiliar with Kegel exercises, and healthcare programs often neglected education on pelvic floor muscle strengthening. Consequently, few women received instruction on Kegel exercises, and among those who did, only a few performed them correctly.
Mokhtari et al. [32]	“Preparation of Protocol for Removing Physical, Psychological, Environmental and Social Barriers to Physical Activity in Pregnant Women”	The study aimed to develop a protocol to address barriers to PA among healthy pregnant women at health centers and hospitals in Isfahan. Pregnant women completed questionnaires on demographics and PA, and solutions to overcome PA barriers were developed using Delphi methodology, a literature review, and expert input. The study identified 61 barriers across physical, psychological, environmental, and social dimensions, with physical and psychological barriers being the most significant. Key facilitators included increasing awareness, education, and massage. The study recommended integrating this protocol into routine prenatal care to enhance PA during pregnancy.
Kiani et al. [52]	“Examining How to Exercise and the Effective Factors on It Based on Perspective of Pregnant Women Referring to Health Centers in Astara City”	This study examined the factors influencing the willingness to exercise during pregnancy in Astara. Data collected via questionnaires (knowledge and the amount of PA) revealed that 45% of pregnant women exercised, while 55% did not. The primary motivator for exercising was recommendations from doctors, midwives, and healthcare personnel. Notably, women in rural areas showed a higher inclination to exercise compared to their urban counterparts. The study highlighted the need for retraining healthcare personnel and improving their knowledge and attitudes towards prenatal exercise to better support and encourage PA during pregnancy.
Abedzadeh et al. [17]	“Knowledge and Performance of Pregnant Women Referring to Shabihkhani Hospital on Exercises during Pregnancy and Postpartum Periods”	This study assessed pregnant women’s knowledge and performance regarding exercise during pregnancy and the postpartum period. Data were collected through a questionnaire covering demographic characteristics and knowledge of, and engagement in, exercise. Results indicated a high percentage of poor knowledge about exercise among pregnant women. The study suggested that increasing awareness of exercise benefits is crucial and recommended regular training sessions, educational pamphlets, and enhanced education for medical and midwifery students to improve women’s attitudes towards exercise during pregnancy.
Abdollahi et al. [45]	“The Effect of Education Based on Self-Efficacy Strategies in Changing Postpartum Physical Activity”	The study examined self-efficacy as a moderator of changes in postpartum PA using the Exercise Self-Efficacy Scale (ESS) and the International Physical Activity Questionnaire (IPAQ). An educational intervention, consisting of two 80-min sessions and practical activities, was designed around self-efficacy strategies. This included posters, fact sheets, a physical activity self-monitoring checklist, and a social network for active mothers on Telegram and the website. Before the intervention, only 15% of mothers had high exercise self-efficacy, and 35% engaged in moderate-to-vigorous physical activity (MVPA). Two months after the intervention, there was a significant increase in physical activity levels. The study suggests that workshops based on self-efficacy and PA strategies should be implemented in health centers for postpartum women.
Heidari Dehui et al. [47]	“Weight Management Challenges in Nulliparous Women Being Overweight or Obese Due to Pregnancy: A Qualitative Study”	This study explored barriers to post-pregnancy weight-management behaviors among obese women using individual, in-depth, semi-structured face-to-face interviews with open-ended questions. The findings revealed that postpartum constraints differed from prenatal ones, particularly in individual and social aspects as well as environmental and health-related factors. To improve healthy behaviors like PA, it is crucial to involve not only relevant stakeholders but also to foster commitment and engagement from women, their families, and their communities.
Pouriayevali et al. [49]	“Mothers’ Views on Mobile Health in Self-Care for Pregnancy: A Step towards Mobile Application Development”	This study aimed to identify the operational needs for a pregnancy self-care application from the perspective of pregnant Iranian women. Through face-to-face semi-structured interviews, the study assessed pregnant women’s perceptions, attitudes, and interests regarding the app’s features. The findings highlighted the importance of managing maternal PA in pregnancy care and emphasized the need for guidance on exercise type, duration, and intensity. The study concluded that a localized pregnancy self-care application is essential to support pregnant women in monitoring health behaviors, preventing obesity, and planning PA programs.

## Data Availability

Data are contained within the article.

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
