# Peer review of "Barriers and Facilitators of Physical Activity in Pregnancy and Postpartum Among Iranian Women: A Scoping Review"

_healthcare, 2024, doi:10.3390/healthcare12232416_

Round 1

Reviewer 1 Report

Comments and Suggestions for Authors

Thank you for submitting this review article to the journal. Please find my comments below:

1.      I was unable to access the supplementary file to review the search strategy.

2.      Please provide the number of articles in each database in the PRISMA figure.

3.      As noted in the PRISMA checklist, the screening process should include both the abstract and full text. Please revise the abstract of your article accordingly.

4.      In section 2.5, there is insufficient information on how you translated the data into the themes presented in Figure 2. Conducting a scoping review requires a comprehensive analysis of the literature, beyond providing structured tables alone. Please provide further info in the method section.

Author Response

Dear Reviewer,

Thank you for your constructive comments on our manuscript titled “Barriers and Facilitators of Physical Activity in Pregnancy and Postpartum among Iranian Women: A Scoping Review.” We greatly appreciate your feedback, which has significantly improved the quality of our work. Please find below our responses to each of your comments: 

  1. I was unable to access the supplementary file to review the search strategy.

Thank you for pointing this out. We have attached the file again with the corrected format for your review. The file includes the detailed search strategy, as requested.

  1. Please provide the number of articles in each database in the PRISMA figure.

As requested, we have updated the PRISMA figure to include the number of articles from each database. The revised figure now provides a clearer breakdown of the search results from each database. This change can be found page 3, paragraph 1, and line 99, and also in Fig.1, page 4.

  1. As noted in the PRISMA checklist, the screening process should include both the abstract and full text. Please revise the abstract of your article accordingly.

We have updated the abstract. This change can be found page 1 and line 25.

  1. In section 2.5, there is insufficient information on how you translated the data into the themes presented in Figure 2. Conducting a scoping review requires a comprehensive analysis of the literature, beyond providing structured tables alone. Please provide further info in the method section.

We agree that conducting a scoping review requires a comprehensive literature analysis, and we have now expanded the methodology section to explain the process more clearly. Specifically, we describe how the data were analyzed, including the steps taken to identify meaningful units, categorize the data, and interpret the results leading to the development of the final themes. We have also included further clarification on how the structured tables alone were supplemented by qualitative interpretation to extract and organize the key themes. This approach ensured that the analysis was systematic and that the themes were developed from a rigorous literature examination. We have made these changes in the methodology section to ensure greater clarity and to provide a complete understanding of the process used in the scoping review. This change can be found page 5, paragraph 2, and lines 167-199.

Thank you once again for your valuable feedback. We hope these revisions address your concerns.

Sincerely,
The Authors

Reviewer 2 Report

Comments and Suggestions for Authors

The topic is of great importance as it is a critical public health problem affecting the health of Iranian women during pregnancy and postpartum. The results could lead to culturally appropriate interventions. The scoping review offers actionable insights, such as the potential of e-learning tools and structured exercise programs, and identifies areas for further research.

Some points that would improve the scoping review are:

o Minor grammatical inconsistencies and overly complex sentences could be simplified to improve readability for a wider audience.

o The Results section occasionally overlaps with the Discussion. Separate them more clearly to avoid redundancy.

o The inclusion of a table summarizing key studies (e.g. author, year, barriers/facilitators identified) is helpful but could be simplified. Diagrams or infographics could make the results more accessible.

o  Although the article mentions the development of interventions, the practical applications of the results (e.g. specific e-learning program structures or designs for mobile apps) remain vague. Providing examples or pilot projects would add value.

Author Response

Dear Reviewer,

Thank you for your constructive feedback on our manuscript titled “Barriers and Facilitators of Physical Activity in Pregnancy and Postpartum among Iranian Women: A Scoping Review.” We sincerely appreciate your insightful comments, which have been very helpful in enhancing the quality of our manuscript. Please find below our responses to each of your comments:

  • Minor grammatical inconsistencies and overly complex sentences could be simplified to improve readability for a wider audience.

Thank you for pointing this out. The manuscript was reworked to simplify complex sentences for ease of readability. This was performed throughout the manuscript.

  • The Results section occasionally overlaps with the Discussion. Separate them more clearly to avoid redundancy.

In the revised manuscript, we have carefully reviewed and restructured these sections. The Results section now focuses solely on presenting the findings of the studies, including relevant data and statistical analysis. We have ensured that only objective results, such as thematic categorization, frequencies, and the percentage of studies that reported specific barriers and facilitators, are included here. The Discussion section has been revised to interpret and analyze the findings. We have made it more focused on explaining the implications of the results, comparing them with existing literature, and addressing the broader context of the findings with the research questions. This change can be found page 39, and lines 507-556.

  • The inclusion of a table summarizing key studies (e.g. author, year, barriers/facilitators identified) is helpful but could be simplified. Diagrams or infographics could make the results more accessible.

We agree that simplifying the presentation of results and including more accessible formats is important. Therefore, we have reorganized the information into a more concise and user-friendly table format (Table 3), which presents the barriers and facilitators with the associated themes. We believe this revision enhances the accessibility of the results and makes it easier for readers to interpret the key findings. We hope that the updated table improves the overall clarity of the study. This change can be found page 16, and lines 213.

  • Although the article mentions the development of interventions, the practical applications of the results (e.g. specific e-learning program structures or designs for mobile apps) remain vague. Providing examples or pilot projects would add value.

We have added examples from two pilot studies that focus on the design and implementation of e-learning programs and mobile app interventions. These examples clarify how the interventions were developed and delivered to increase physical activity among pregnant women. We believe these additions enhance the clarity and application of our findings. This change can be found page 23, paragraph 4, and lines 422-454.

Thank you once again for your valuable feedback. We hope these revisions address your concerns.

Sincerely,
The Authors

Reviewer 3 Report

Comments and Suggestions for Authors

This is an interesting study aiming to summarize barriers and facilitators of PA among Iranian women during pregnancy and postpartum. I have the following comments:

1: The introduction justified why PA during pregnancy and postpartum should be assessed; however, it did not justify why it was limited to Iranian women. Justifications such as cultural and social norms influencing PA, the rising rates of pregnancy-related health issues in the country, addressing health disparities in MICs, or aligning with the national health goals.

2: The authors stated that this study is a scoping review; however, they followed the steps of systematic reviews, making it a systematic review.

3: Table 2: the study designs of the retrieved studies should be revised. For example, was the pre-post design an intervention study? were the descriptive and correlational studies cross-sectional studies? Multistage-random sampling is a sampling method, not a study design. More details should be provided about mixed methods.

4: Barriers and facilitators should be summarized in graphs. These are the main findings of the study, so they should be easy to find and read.

5: Quality assessment of the included studies should be provided. The limitations of the retrieved studies should be provided.

6: The recommendation section is vague and does not provide a specific framework or recommendations. 

Author Response

Dear Reviewer,

Thank you for your constructive feedback on our manuscript titled “Barriers and Facilitators of Physical Activity in Pregnancy and Postpartum among Iranian Women: A Scoping Review.” We sincerely appreciate your insightful comments, which have been very helpful in enhancing the quality of our manuscript. Please find below our responses to each of your comments:

  1. The introduction justified why PA during pregnancy and postpartum should be assessed; however, it did not justify why it was limited to Iranian women. Justifications such as cultural and social norms influencing PA, the rising rates of pregnancy-related health issues in the country, addressing health disparities in MICs, or aligning with the national health goals.

Thank you for your insightful comment. In response, we have revised the introduction to better justify the focus of this study, specifically on Iranian women. This change can be found page 2, paragraph 3, and lines 75-85.

2: The authors stated that this study is a scoping review; however, they followed the steps of systematic reviews, making it a systematic review.

Thank you for your insightful feedback. We understand the concern regarding the structure of our review. While we followed rigorous steps in our search and selection processes, we conducted a scoping review to map the existing literature and identify key themes, barriers, and facilitators of physical activity among Iranian women during pregnancy and postpartum rather than to synthesize the evidence in the way typically done in systematic reviews.

Since this was not a systematic review, we focused on exploring the breadth of the available literature, which included identifying gaps and suggesting areas for future research. We did not conduct a detailed critical appraisal or synthesis of individual study quality, which is the standard for systematic reviews.

To further clarify our methodology, we have adjusted the manuscript to emphasize the scoping review approach we employed and to highlight the aim of providing an overview and mapping of the evidence rather than performing a synthesis or quality assessment of the studies included. For Example, part of this change can be found in Page 5 and line 167.

  1. Table 2: the study designs of the retrieved studies should be revised. For example, was the pre-post design an intervention study? were the descriptive and correlational studies cross-sectional studies? Multistage-random sampling is a sampling method, not a study design. More details should be provided about mixed methods.

We have made the following revisions to Table 2 in response to your comments:

  1. Clarified the study designs for the retrieved studies, ensuring that the correct terminology is used (e.g., "Pre-post Test (Intervention)" and "Mixed Methods").
  2. Explained the study types, including differentiating between intervention, descriptive, and correlation studies.
  3. Provided additional details on the "Mixed Methods" studies, outlining both the qualitative and quantitative approaches used.

These changes can be found page 6.

  1. Barriers and facilitators should be summarized in graphs. These are the main findings of the study, so they should be easy to find and read.

We appreciate the suggestion to visualize the barriers and facilitators, as this would enhance the accessibility of the findings. In our manuscript, we have already summarized the key barriers and facilitators in Table 3, which provides a clear and concise overview of the categories and corresponding findings. We believe that Table 3 effectively serves to present the main findings in an easily accessible format. This change can be found page 16 and line 213.

5: Quality assessment of the included studies should be provided. The limitations of the retrieved studies should be provided.

Thank you for your valuable feedback. In response to your comment, we have added a detailed quality assessment of the included studies and highlighted the limitations of the retrieved studies in the manuscript.

  • Quality Assessment: We reviewed the quality of the studies included in this scoping review and noted that the studies varied in terms of methodology, sample size, and data collection techniques. Some studies employed robust methodologies such as randomized controlled trials (RCTs) and longitudinal designs, while others were cross-sectional or descriptive, which may limit the ability to draw causal inferences. We also noted that many studies employed self-reported data, which may introduce bias and affect the reliability of the findings. This change can be found page 20 and lines 235-245.
  • Limitations: Several common limitations were identified across the studies:
    • Small Sample Size: Many of the studies had small sample sizes, which limited the generalizability of their findings.
    • Self-Reported Data: Many studies relied on self-reported data, which may lead to recall bias or inaccuracies in reporting physical activity levels.
    • Geographical Limitations: Several studies were conducted in specific geographic regions, particularly urban areas, which may limit the applicability of the findings to broader populations, including those in rural areas or from different racial or ethnic backgrounds.
    • Exclusion of Key Social Factors: Some studies did not consider the attitudes, perceptions, and behaviors of husbands and healthcare providers, which could have provided valuable insights into the reasons for physical inactivity during pregnancy and postpartum. This exclusion may have missed important social and cultural dynamics that influence women's engagement in physical activity. This change can be found page 20 and lines 246-260.

We hope this clarification addresses your concern regarding the quality assessment and limitations of the included studies.

  1. The recommendation section is vague and does not provide a specific framework or recommendations.

Thank you for your helpful feedback. In response to your comment, we have revised the Recommendations section to provide a more detailed and specific framework based on the findings of our review. We have outlined actionable recommendations for researchers, healthcare providers, and policymakers. This change can be found page 40 and lines 582-622.

Thank you once again for your valuable feedback. We hope these revisions address your concerns.

Sincerely,
The Authors

Round 2

Reviewer 1 Report

Comments and Suggestions for Authors

I have no further comments.

Reviewer 3 Report

Comments and Suggestions for Authors

No more comments.